# Structure and electrochromism of two-dimensional octahedral molecular sieve h'-WO$_3$

Julie Besnardiere[1], Binghua Ma [1], Almudena Torres-Pardo [2], Gilles Wallez [3], Houria Kabbour [4], José M. González-Calbet [2,5], Hans Jürgen Von Bardeleben[6], Benoit Fleury [7], Valérie Buissette [8], Clément Sanchez [1], Thierry Le Mercier [8], Sophie Cassaignon [1] & David Portehault [1]

Octahedral molecular sieves (OMS) are built of transition metal-oxygen octahedra that delimit sub-nanoscale cavities. Compared to other microporous solids, OMS exhibit larger versatility in properties, provided by various redox states and magnetic behaviors of transition metals. Hence, OMS offer opportunities in electrochemical energy harnessing devices, including batteries, electrochemical capacitors and electrochromic systems, provided two conditions are met: fast exchange of ions in the micropores and stability upon exchange. Here we unveil a novel OMS hexagonal polymorph of tungsten oxide called **h'-WO$_3$**, built of (WO$_6$)$_6$ tunnel cavities. **h'-WO$_3$** is prepared by a one-step soft chemistry aqueous route leading to the hydrogen bronze **h'-H$_{0.07}$WO$_3$**. Gentle heating results in **h'-WO$_3$** with framework retention. The material exhibits an unusual combination of 1-dimensional crystal structure and 2-dimensional nanostructure that enhances and fastens proton (de)insertion for stable electrochromic devices. This discovery paves the way to a new family of mixed valence functional materials with tunable behaviors.

[1] Sorbonne Université, CNRS, Collège de France, PSL Research University, Laboratoire Chimie de la Matière Condensée de Paris, LCMCP, 4 Place Jussieu, F-75005 Paris, France. [2] Departamento de Química Inorgánica, Facultad de Químicas, Universidad Complutense, 28040 Madrid, Spain. [3] PSL Research University, Chimie ParisTech, CNRS, Institut de Recherche de Chimie de Paris, 11 rue Pierre et Marie Curie, 75005 Paris, France. [4] Univ. Lille, CNRS, ENSCL, Centrale Lille, Univ. Artois, UMR 8181-UCCS-Unité de Catalyse et de Chimie du Solide, F-59000 Lille, France. [5] Centro Nacional de Microscopía Electrónica, Universidad Complutense, 28040 Madrid, Spain. [6] Sorbonne Université, CNRS, Institut des Nanosciences de Paris, INSP, 4 Place Jussieu, F-75005 Paris, France. [7] Sorbonne Université, CNRS, Institut Parisien de Chimie Moléculaire, IPCM, F-75005 Paris, France. [8] Solvay, Centre de Recherches d'Aubervilliers, 52 rue de la Haie-Coq, 93308 Aubervilliers Cedex, France. These authors contributed equally: Julie Besnardiere, Binghua Ma. Correspondence and requests for materials should be addressed to D.P. (email: david.portehault@sorbonne-universite.fr)

Molecular sieves (MS) are microporous solids involved since decades in industrial processes such as purification, separation, and petroleum refining, among others[1]. Although most MS including microporous carbons and zeolites are based on the arrangement of four-coordinated C, Al, and Si tetrahedral units, scarcer inorganic solids arise from the organization of $MO_6$ octahedra. These materials were coined as octahedral MS (OMS) by S. L. Suib[2]. Because the six-coordinated cation M is a transition metal, typically manganese[2–4] and less frequently vanadium[5,6], tungsten[7,8], titanium[9], or niobium[10], OMS possess greater versatility than common tetrahedra-based microporous solids in terms of redox[3], magnetic[11] properties and substitution[11–13] abilities, which open a realm of opportunities in catalysis[3,12,13], energy storage[3,14], sensing[15], information technologies[15], and smart systems[16]. Very often, OMS rely on ternary or more complex compositions including guest cations (proton, alkali, and alkali earth ions) or water molecules into micropores. These species act as templates that cannot be totally eliminated without collapsing the structure[2–6]. Tungsten OMS are special in this respect since the W-O framework of h-$WO_3$[7] or pyrochlore-$WO_3$[8] is maintained after removal of guests and concomitant oxidation of charge compensating $W^V$ into $W^{VI}$, yielding stoichiometric $WO_3$ compounds with micropores, contrary to the other $WO_3$ polymorphs[17]. In this article, we expand the $WO_3$ family by using aqueous chemistry to unveil a new tungsten hydrogen bronze **h'-$H_{0.07}WO_3$** with a novel W-O framework and promising electrochemical properties. Protons can be reversibly extracted to yield a novel guest-free, OMS stoichiometric binary tungsten oxide **h'-$WO_3$**, with high chemical and thermal stability.

Many of the properties of tungsten oxides for gas sensors[17,18], electrochromic devices[19–21], supercapacitors[22], batteries[23–28], photocatalysts[29], water splitting devices[30], and solar cells[31] rely on the ability of $WO_3$ compounds to insert cations and therefore to form bronzes $A_xWO_3$ (A being a cation such as proton, alkali ion, or ammonium), concomitant with partial reduction of $W^{VI}$ centers to $W^V$.

We show hereafter that a chimie douce route yields anisotropic two-dimensional nanostructures of the new **h'-$WO_3$** framework that result in preferential orientation of the micropores for easy and fast ion insertion/deinsertion, thus leading to a stable and efficient electrochromic material in aqueous medium.

## Results and discussion

**Synthesis procedure.** The new tungsten oxide powder was prepared in water under soft conditions. Briefly, the pH of a $Na_2WO_4·2H_2O$ and hydrazine $N_2H_4$ solution was adjusted to a value of 0.6, close to the isoelectric point of tungsten oxides (~ 1.5)[32] and hydrous tungsten oxides (~ 0.5)[33] leading to a beige amorphous precipitate. The suspension was aged at room temperature for 14 h and then at 95 °C for 3 days or at 120 °C under autogenic pressure for 12 h. A deep-blue powder characteristic of mixed valence $W^V/W^{VI}$ tungsten oxides[17] was recovered after washing and drying at 40 °C under vacuum.

**Crystal structure and nanostructure.** Scanning electron microscopy (SEM) and transmission electron microscopy (TEM) (Fig. 1a, b) show that the powder is exclusively made of thin nanoplatelets with diameter between 20 and 80 nm and thickness of 3–10 nm (Fig. 1b, statistical measurements in Supplementary Figure 1). High-resolution TEM (HRTEM) and selected area electron diffraction (SAED) (Fig. 1b) reveal a hexagonal tiling with a characteristic distance of 8.7 Å, which could not be attributed to any of the known tungsten oxides or bronzes. As HRTEM images and SAED patterns, the powder X-ray diffraction (XRD) pattern (Supplementary Figure 2) cannot be indexed according to known phases and supports the discovery of a new W-O-based structure.

The XRD pattern has been indexed in a hexagonal cell (space group P6/mmm) with lattice parameters $a = 9.997$ Å and $c = 3.921$ Å (Supplementary Table 1, crystallographic information file (CIF) and CheckCIF file as Supplementary Data 1, Supplementary Data 2, respectively). The $c$ value corresponds to one $WO_6$ octahedron layer, already observed in the classical **h-$WO_3$** structure. The cell volume of 339.3 Å³ can accommodate six $WO_3$ formula units with a theoretical density of 6.81 in the range of those for known $WO_3$ polymorphs and bronzes, especially **h-$WO_3$** (6.49)[7]. The anisotropy of the single crystal particles and the presence of stacking faults discussed below were taken into account for Rietveld refinement (Fig. 1e and SI)[34]. Projections of the new hexagonal structure along the **c** axis and in the [110] direction are shown in Figs. 1f, g, respectively. The structure is based on an arrangement of $WO_6$ octahedra sharing corners in $(WO_6)_6$ wheels, which are stacked along the **c** axis to yield tunnels. These tunnels exhibit an internal diameter after subtraction of oxygen radii of ca. 4.8 Å, typical of OMS solids. The tunnels are connected together by octahedra corners, which open additional $(WO_6)_4$ and $(WO_6)_3$ tunnels. Further investigation of the structure was performed by scanning TEM. Because the sample is very sensitive to beam damage, STEM imaging was performed in an aberration-corrected microscope enabling operation at low acceleration voltages (80 kV) and low dose with high spatial resolution. Both high angle annular dark field (HAADF) and annular bright field (ABF) detections were used (Fig. 1), with sensitivity to heavy and light elements, respectively. The structure derived from powder XRD is consistent with STEM data (Fig. 1c, inset).

The nearly perfect superimposition of the XRD-resolved structure over the atomically resolved HAADF-STEM micrograph (Fig. 1c) confirms the accuracy of our refinement. However, a range of crystal defects is observed by STEM (Fig. 2a), which encompasses (1) 30°-rotation of $(WO_6)_6$ wheels around the **c** axis (Fig. 2b, d, e), (2) antiphase boundaries (Fig. 2c), and (3) five-membered rings (Fig. 2a) where one $WO_6$ unit has been stripped off $(WO_6)_6$ wheels. 30°-rotations result in strong electron density residuals on the Fourier-difference maps when Rietveld refinement is performed only with the ideal structure. Hence, these rotations have been taken into account for structure refinement by considering a solid solution of the regular structure and the faulted one (see SI). The reliability was significantly improved and reached a satisfactory level (SI). The proportion of 30°-rotation defects according to Rietveld refinement corresponds to 14% of the six-members tunnels, in agreement with TEM observations (ca. 10%). The size of the six-members cavity is slightly reduced in the presence of the 30°-rotations (internal diameter after subtraction of oxygen radii changing from 4.8 Å to 4.5 Å in the ideal and rotated wheels, respectively).

The resolved W-O skeleton exhibits little electron density residues in the hexagonal cavities, which reveal the presence of additional atoms. Indeed, the blue hue and the band gap measured by UV-vis spectroscopy of about 2.9 eV are consistent with other $WO_3$ bronzes[17] and are typical of mixed valence W oxides that exhibit $W^V/W^{VI}$ intervalence transitions at wavelengths above 400 nm (Supplementary Figure 3)[19]. The mixed valence is confirmed by electron spin resonance (ESR, Supplementary Figures 4-8). Note that ESR suggests important spin coupling that is currently under investigation (Supplementary Figures 6-7). Small cations are inserted in the $WO_3$ framework to ensure charge balance. According to the synthesis protocol, $H^+$, $Na^+$ from the tungstate salt and $NH_4^+$ from hydrazine are the only candidates. $NH_4^+$ is absent because nitrogen is not observed by energy dispersive X-ray spectroscopy (EDS) and replacement of hydrazine with amine-free glucose or sodium ascorbate yields the same solid. Exchange of the inserted ions by an excess of $Na^+$

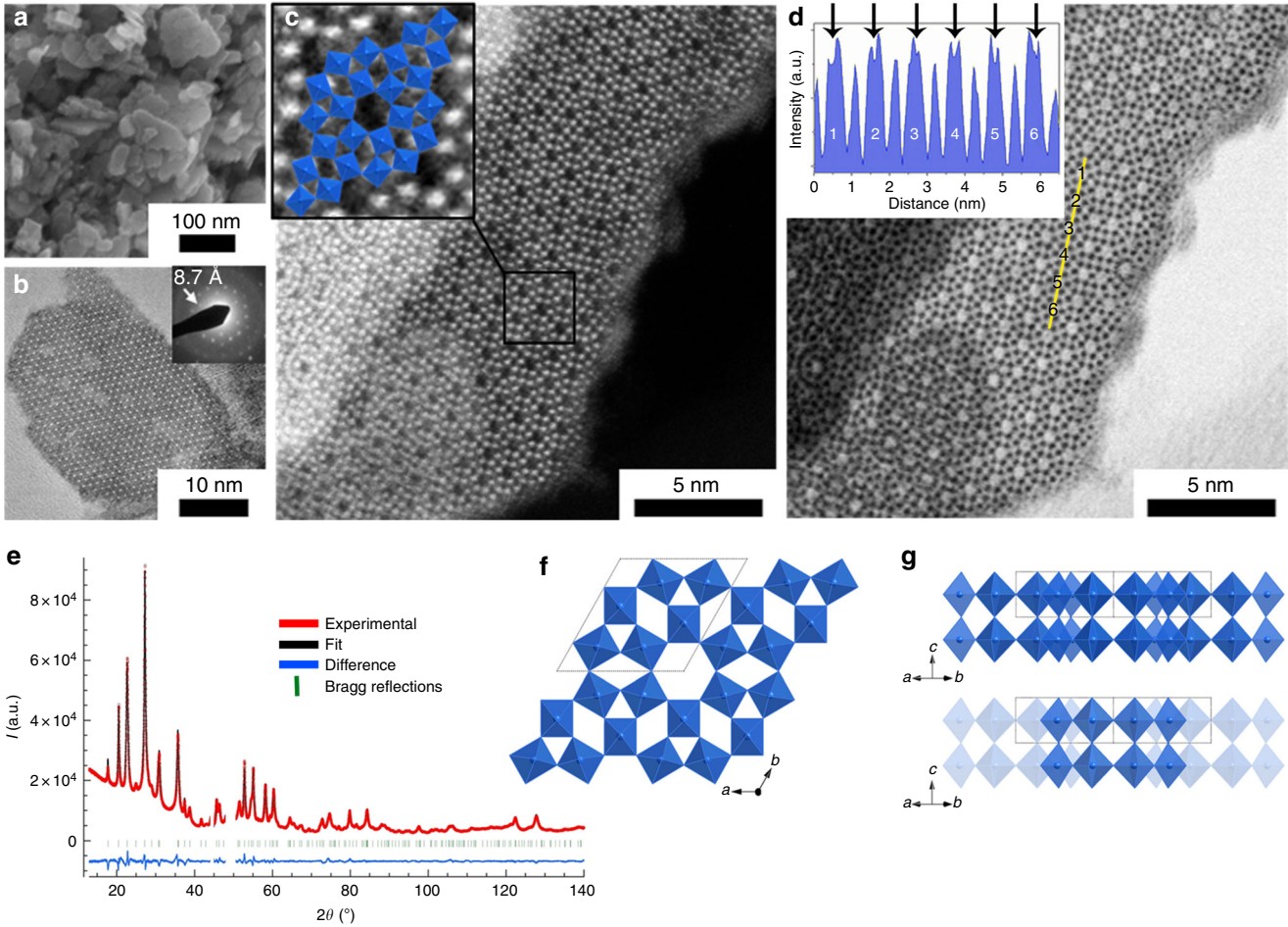

**Fig. 1** Structure and nanostructure of the **h'-WO₃** framework. The study has been performed on the bronze recovered directly after aqueous synthesis. **a** SEM and **b** HRTEM images showing nanosized platelets. The inset in (**b**) shows a typical hexagonal SAED pattern. **c** HAADF-STEM and **d** corresponding ABF-STEM micrographs showing the arrangement of tungsten octahedra (blue, in inset of (**c**)). **e** Rietveld refined powder XRD pattern (Cu $K_\alpha$). Projections of the structure along the (**f**) c axis and (**g**) the [110] direction. Bottom of (**g**) shows the structure with depth fading to highlight square channels along the [110] direction. The inset in (**d**) shows contrast variations along the yellow line. Each large increase in intensity corresponds to each numbered $(WO_6)_6$ tunnel along the line. Black arrows show depressions of intensity in each tunnel, highlighting inserted cations

in water (see Methods) yields fast acidification. Accordingly, the $H^+$/W ratio is evaluated to 8 at. %. The additional presence of $Na^+$ is unlikely as sodium could not be detected by EDS (Supplementary Figure 9). This is confirmed by X-ray photo-electron spectroscopy (XPS, Supplementary Figure 10) that yields a $W^{5+}$/W ratio of ca. 7 ± 3 at. %, in agreement with the proton content. Hence, sodium ions do not participate to charge balance and the bronze obtained after aqueous synthesis is a hydrogen bronze of overall composition **h'-$H_{0.07 \pm 0.03}$WO₃**. The presence of protons in the tunnels is confirmed by the STEM-ABF images (Figs. 1d and 2e) that show a dark contrast in all the $(WO_6)_6$ tunnels (Fig. 1d, inset), whereas HAADF (Fig. 1c) does not exhibit such features, thus evidencing a light element: hydrogen atoms[35]. Note that some of the $(WO_6)_4$ channels show also a dark contrast in ABF detection (Supplementary Figure 11), which would suggest that some protons also occupy part of these four-members tunnels. Calcination of this **h'-$H_{0.07}$WO₃** bronze under air at 100 °C readily yields a white, fully oxidized, material with identical XRD pattern (Fig. 3a, equation 1). Thus, the new W-O framework is maintained, yielding a stoichiometric white oxide we note **h'-WO₃**. Charge compensating cations present in the bronze can then be eliminated through a topotactic reaction by soft annealing, which supports the presence of protons that can evolve as water molecules. Hence, the as-obtained solid **h'-**

**$H_{0.07}$WO₃** is a hydrogen bronze of **h'-WO₃**. Thermal gravimetric analysis (TGA) combined with differential temperature analysis (DTA) and temperature-dependent XRD (Fig. 3) shows that the structure is preserved up to 425 °C under air. Such a thermal stability is higher than classical **h-WO₃** transforming at 300 °C and interesting for many applications. The transition from the bronze to the stoichiometric oxide is reversible (Fig. 3a, equation 2): upon exposure to air and light at room temperature for about 10 min, the material recovers its initial blue color. Interestingly, under air but in the dark, the white stoichiometric **h'-WO₃** is stable, thus highlighting the photosensitivity of this new solid.

**Optical and electronic properties**. The wide UV-visible absorption band for the bronze **h'-$H_x$WO₃** at wavelengths above 400 nm (Supplementary Figure 3) is related to intervalence transitions between $W^{5+}$ and $W^{6+}$ that correspond to optically activated electron transfer associated with the presence of polarons[19,36]. According to previous reports on other WO₃ polymorphs, the x value of ca. 0.07 should be too low to enable overlapping of the polaronic wave functions and the emergence of a metallic state[36-38]. Hence, the bronze is expected as a semi-conductor and the band gap evaluated by UV-visible spectroscopy from the lower wavelength region should be similar in the oxidized state. To get deeper insight into the electronic properties,

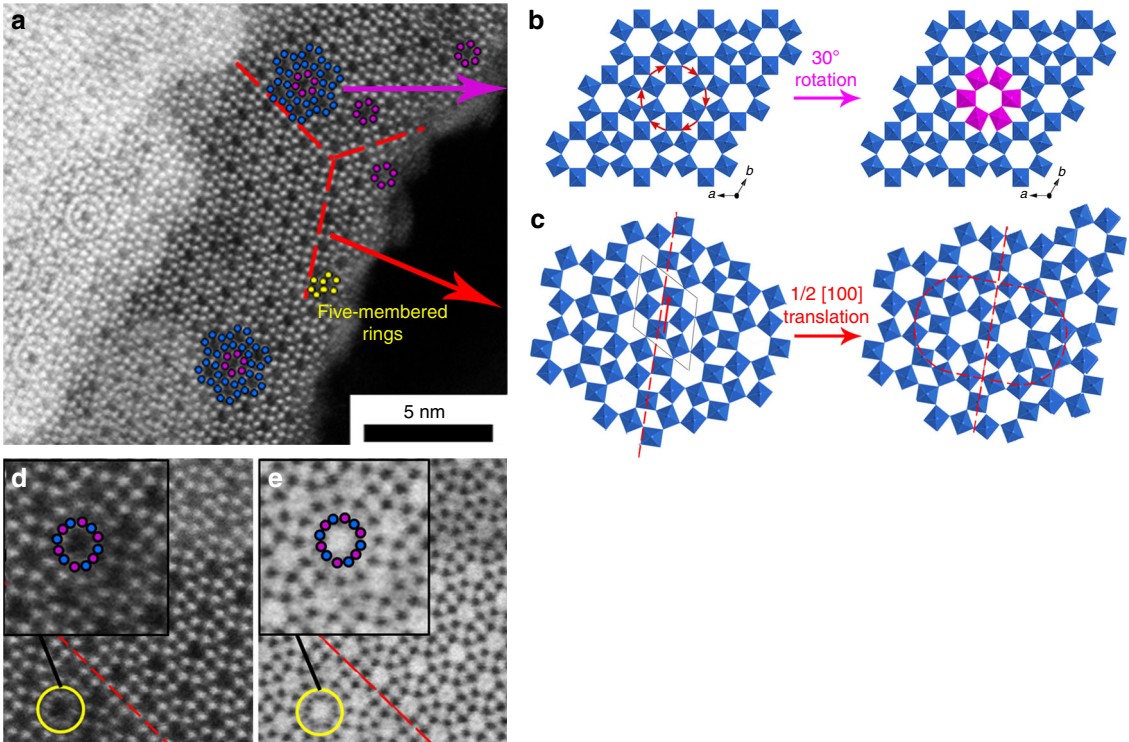

**Fig. 2** Crystal defects in the h'-WO₃ framework. The study has been performed on the bronze recovered directly after aqueous synthesis. **a** STEM-HAADF micrograph showing in yellow five-membered $(WO_6)_5$ rings where one octahedra has been removed from a $(WO_6)_6$ wheel, **b** 30°-rotation of $(WO_6)_6$ wheels around the **c** axis (purple), **c** antiphase boundaries showing displacement of the structure by a vector ½ [100] along (100) directions (red dashed lines). **d** STEM-HAADF and **e** corresponding STEM-ABF images of superimposed $(WO_6)_6$ rings rotated by 30° with respect to each other

the density of states (DOS) of **h'-WO₃** was calculated by first principle density functional theory (DFT). In order to take into account the influence of the 30°-$(WO_6)_6$ rotation defects, we have also assessed the electronic structure of a phase in which all $(WO_6)_6$ wheels are rotated (denominated ***rotated h'*** hereafter, Fig. 4). For comparison, the DOS of already known hexagonal **h-WO₃** and monoclinic **m-WO₃** polymorphs have been also calculated (Fig. 4). **h-WO₃** also contains $(WO_6)_6$ and $(WO_6)_3$ tunnels, but does not exhibit $(WO_6)_4$ tunnels.

The new **h'** phase and its derivate show similar total, O 2p and W 5d-projected DOS that are also close to those of different WO₃ polymorphs (Fig. 4). All phases are semiconductors, with valence and conduction bands dominated by O 2p and W 5d orbitals, respectively, which is commonly reported for WO₃[39–41] and WO₃ hydrate phases[42]. The band gaps evaluated by DFT are 0.59, 0.65, 0.52, and 1.22 eV for the **h'**, **rotated h'**, **h**, and **m** polymorphs. These values are consistent with previous reports on **h-WO₃** and **m-WO₃** using DFT-PBE (Perdew-Burke-Ernzerhof) calculations[39–41] that commonly underestimate band gaps (see SI). Hence, the calculated values are lower than the experimental ones reported for the **h** and **m** polymorphs (2.5–3.0 eV)[17] and measured above for the **h'** bronze (2.9 eV). The contribution of each **h'** and **rotated h'** networks to the actual total DOS was roughly estimated from the proportion of each network obtained from XRD-based structure refinement (Supplementary Figure 12). The band gap is dictated by the smaller gap network **h'**. From the total energies of the four polytypes calculated, the relative stability order is **rotated h'-WO₃ < h-WO₃ < h'-WO₃ < m-WO₃**. As expected from the literature, the most stable phase at low temperature is the monoclinic one. If **h'-WO₃** is slightly more stable than the standard hexagonal phase, the **rotated h'** network is less stable, which might explain its smaller (~15 %) contribution to the title compound.

We have then introduced hydrogen in the $(WO_6)_6$ channels of our model in two different configurations (Supplementary Figures 13 and 14): either in the center of the tunnels or in the vicinity of oxygen at the edge of the channels. Protons are probably located into the $(WO_6)_6$ channels, but may also be present in the $(WO_6)_4$ channels, as already reported for lithium in the $(WO_6)_4$ openings of **h-WO₃**[43]. More extensive calculations would be needed to examine the precise position that is favored by protons, but the conclusions drawn from our simple models should still be qualitatively valid. DFT (GGA-PBE) study of the H inserted **h'-WO₃** model with OH moieties in the tunnels led to an electronic structure with close features to similar calculations on other polymorphs[44] and hydroxides (see SI). N. Bondarenko et al.[45] reported in detail, with DFT + U approach, various configurations/sites of localized charge in the polaron model that give rise to light absorption in Li-doped γ-WO₃ and γ-WO₃₋ₓ. The coloration mechanism of **h'-WO₃** upon proton insertion most probably relies on similar effects.

**Electrochromic properties**. The nanostructure of these new phases **h'-WO₃** and **h'-H₀.₀₇WO₃** is peculiar in the sense that it is obtained as two-dimensional (2D) nanostructures with (001) basal faces, which are perpendicular to the six-membered channels. Such a crystal habit is unusual for tunnel-based OMS solids, which grow preferentially as nanorods or nanowires along the tunnels axis[46,47]. In the case of **h-WO₃** and its bronzes, additives are necessary to obtain platelets[37,48,49] and to our knowledge, the particles' short dimension was never reported along the **c** axis. If the origin of this preferential growth mode requires further investigations, the consequence is that **h'-WO₃** and its bronze show a new structure but also an original structure–morphology relationship. The 2D nanostructure is combined with an original crystallographic orientation that provides optimal access to the

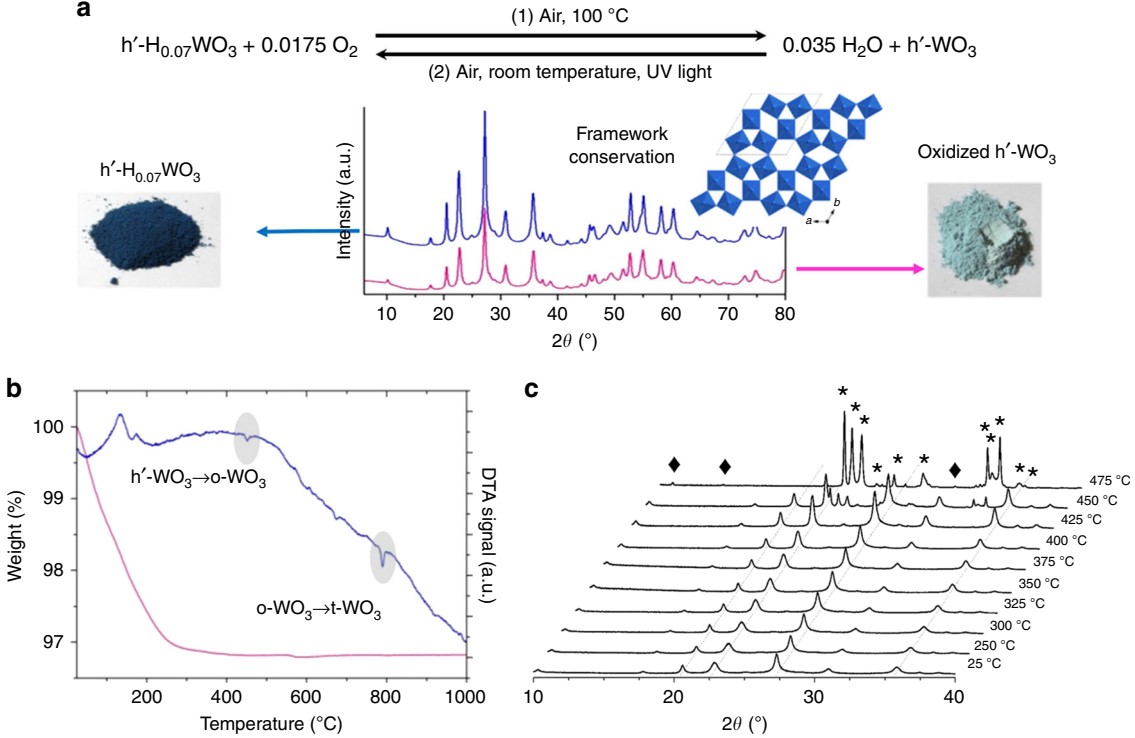

**Fig. 3** Thermal stability of the new **h'-WO₃** framework. **a** XRD patterns and corresponding photographs of **h'-H₀.₀₇WO₃** (before thermal treatment) and **h'-WO₃** (after heating at 100 °C under air) powders. Note that the **h'-WO₃** appears blueish as it is shortly exposed to air in order to take the photograph, so that it already starts to reduce again. **b** TGA and DTA traces under air of the hydrogen bronze. **c** Powder XRD patterns of the heat-treated bronze as a function of the annealing temperature. Stars indicate peaks of the orthorhombic WO₃ structure, lozenges indicate peaks probably due to additional distortions in the structure. The **h'-WO₃** framework is maintained up to 425 °C where it transforms into orthorhombic WO₃ (o-WO₃), then into tetragonal WO₃ (t-WO₃) at 780 °C (**b**)

tunnels and consequently to the microporosity of the material with short diffusion lengths for the insertion of foreign cations. $N_2$ sorption revealed a specific surface area of 55 $m^2 g^{-1}$ consistent with the dimensions deduced from microscopy (Supplementary Figure 15). This value is among the highest surface areas reported for pure WO₃ and bronzes, especially mesoporous c-WO₃₋ₓ and t-WO₃[22]. A priori, **h'-WO₃** with additional four-members tunnels features a more open structure than known W-O-based compounds. It also exhibits a novel texture with easy access to the tunnels, combined with nanostructuration for extensive exchange with the neighboring medium. This unprecedented combination pleads in favor of fast insertion and diffusion of cations in the structure. Swift insertion/deinsertion can be combined with the white-to-blue color change upon evolution from the stoichiometric oxide to the bronze and with the high stability of the **h'-WO₃** framework in acidic electrolytes, in order to build swift electrochromic systems. Tungsten oxides are efficient electrochromic materials[17,21,50,51], but due to its crystal structure and texture, **h'-WO₃** and its proton bronze might yield even faster switch than known WO₃ polymorphs[16,17,19–21,43,47,52–55]. This prompted us to build transparent electrodes for evaluation of aqueous electrochromic devices. The nanoscaled platelets were readily processed as transparent films by spin coating on conductive fluorine-doped tin oxide (FTO) substrates (Fig. 5a, b, SI). The film thickness was evaluated to 80–100 nm (Supplementary Figure 16), which corresponds to approximately 8–20 stacked nanoplatelets. For comparison, films made of classical **h-WO₃** nanoparticles (Supplementary Figure 17) with specific surface area similar to **h'-WO₃** were deposited. The electrochromic properties were then evaluated by using a standard three-electrodes cell. Because of the high stability of **h'-WO₃**, we have focused on an acidic ($H_2SO_4$ 0.1 mol L⁻¹) aqueous electrolyte that imposes stringent requirements vs. possible

dissolution of the electrode materials and in which electrochromic tungsten oxides usually show limited stability[17]. The electrochemical potentials used for coloration/discoloration switch were chosen to ensure reduction and oxidation reactions according to the cyclic voltammograms (Supplementary Figure 18). First, color switching was qualitatively evidenced by applying a square function between 0.8 and −0.8 V vs. AgCl/Ag (Fig. 5a, b) at high frequency of 0.2 Hz. Reversible oscillations between deep-blue and colorless states are observed for **h'-WO₃** (Supplementary Movie 1). These color changes suggest that the amount of protons into the **h'-HₓWO₃** bronze can be controlled by electrochemical (de)insertion. Color variations with classical **h-WO₃** are less pronounced. Bronzes of **h-WO₃** with the same particle size give results similar to **h-WO₃**. According to the $I(t)$ curves (Fig. 5a, b), near-equilibrium conditions are reached upon bleaching (positive bias for oxidation) for both compounds and upon coloration (negative bias for reduction) for classical **h-WO₃**. On the contrary, the equilibrium state is not reached in colored **h'-WO₃** but proton insertion is already sufficient to ensure important absorbance switch. Quantitative assessment was then performed. Electrochemical measurements were coupled with in situ UV-visible measurements at a wavelength of 700 nm (Supplementary Figure 19) by using as reference similar FTO electrodes, without deposited tungsten oxide. Conditions closer to equilibrium were sought by applying smaller voltage steps and lower frequency of ±0.2 V vs. AgCl/Ag and 0.009 Hz, respectively (Fig. 5c). For comparison, a **h-WO₃** film was studied in the same way (Fig. 5d, e). After an induction period of two cycles, absorbance changes are stabilized (Fig. 5c–e). $I(t)$ (Fig. 5c, d, top) and absorbance $(t)$ (Fig. 5c, d, bottom) curves show that steady-states are reached at each potential plateau. The transmittance of the **h'-WO₃** film shows strong changes compared with **h-WO₃**, varying from 95% (absorbance of 0.05 in the bleached state) to 74%

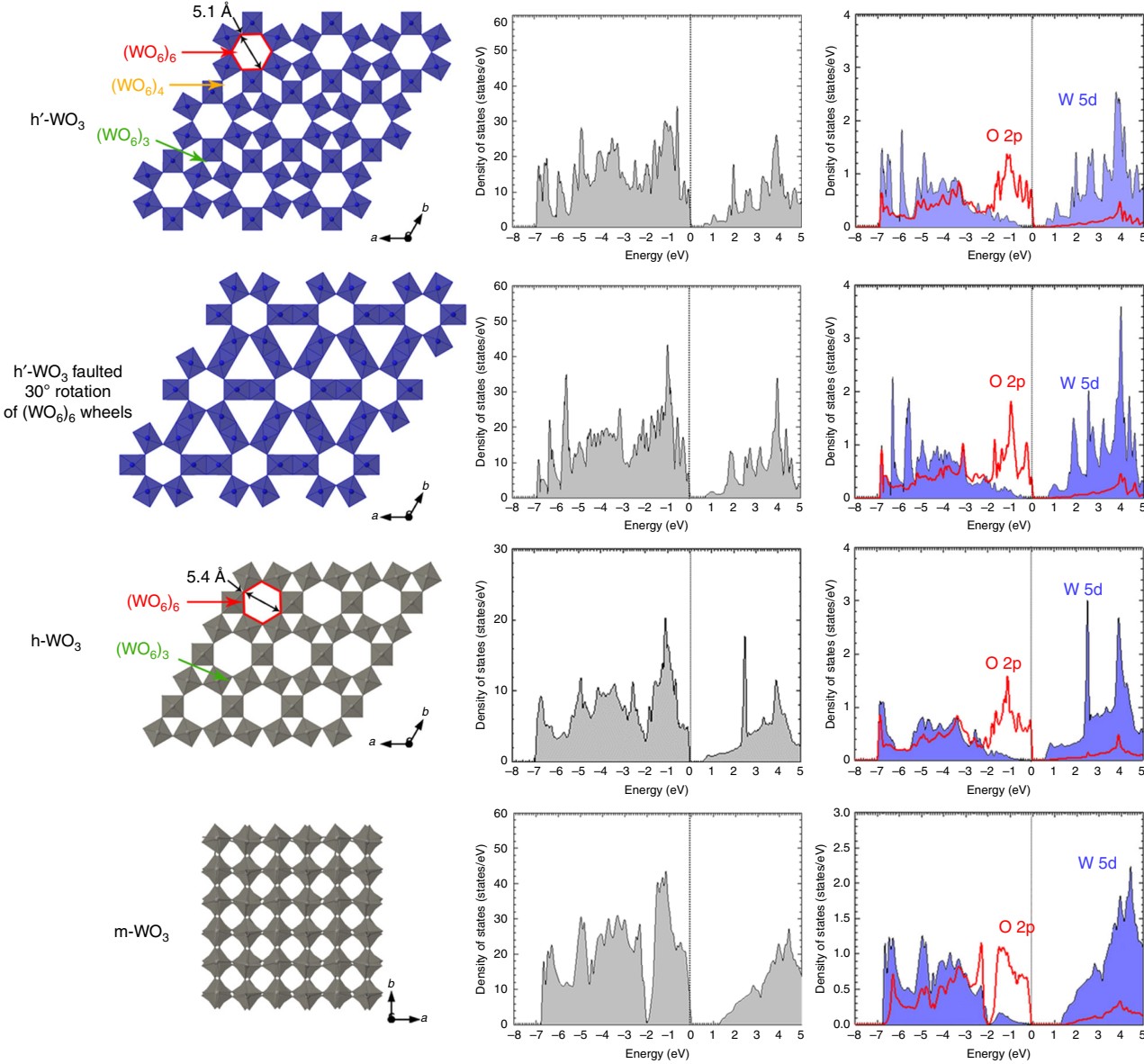

**Fig. 4** Electronic structure of **h'-WO₃**, hexagonal **h-WO₃**, and monoclinic **m-WO₃**. The 30-(WO₆)₆ rotation defects have been taken into account by considering the second phase used for Rietveld refinement, where all (WO₆)₆ wheels are rotated (rotated h'). For hexagonal phases, the diameter of the (WO₆)₆ channels is specified from oxygen ion centers. After subtraction of oxygen radii, the internal channel diameters are ca. 4.8 and 5.1 Å for **h'-WO₃** and **h-WO₃**, respectively. The calculated total densities of states (DOS) are shown for each phase

(absorbance of 0.26 in the colored state) within 13 s. This represents a much larger magnitude than for **h-WO₃** in the same time (from 91 to 88% in 13 s) and other tungsten oxides in the same electrolyte and redox conditions[17]. One can also expect a much deeper and faster switch if higher voltage steps are applied[17,42]. In our electrochemical conditions, the coloration efficiency (CE) at the fourth cycle was evaluated at 53 cm² C⁻¹ and 11 cm² C⁻¹ for **h'-WO₃** and **h-WO₃**, respectively. The relatively high CE value compared with those reported in the literature[17,42] for H⁺ intercalation and the large switching rates highlight the superior optical switch of the new hexagonal framework. Besides, the cycling stability over 150 cycles (16,500 s, Fig. 5e and Supplementary Figures 20 and 21) is excellent. The superior performances of the **h'-WO₃** framework vs. classical **h-WO₃** in terms of response time and extent of coloration, hence of proton (de)insertion, may be ascribed to two features. First, additional four-members (WO₆)₄ tunnels in the **h'** phase can accommodate protons as suggested by STEM-ABF images

(Supplementary Figure 11) and contribute to an increase in the coloration. Second, the original crystallographic orientation of the nanoplatelets provides optimal access to the tunnels of the **h'** phase, whereas the channels of **h-WO₃** are accessible only from the tips of the nanorods (Supplementary Figure 17), with longer diffusion lengths for H⁺ (de)insertion, hence smaller (de)insertion rate. Overall, the novel WO₃ polymorph **h'-WO₃** appears as an excellent candidate for the design of fast electrochromic devices.

**Specificities of h'-WO₃ and its proton bronze.** A new polymorph of WO₃, **h'-WO₃**, and its proton bronze **h'-H₀.₀₇WO₃** have been discovered through a chimie douce pathway leading to the precipitation of the bronze, which could be oxidized in the stoichiometric framework by smooth thermal treatment. The new structure relies on an original hexagonal structure sharing three- and six-octahedra tunnels with the classical hexagonal **h-WO₃** structure, and

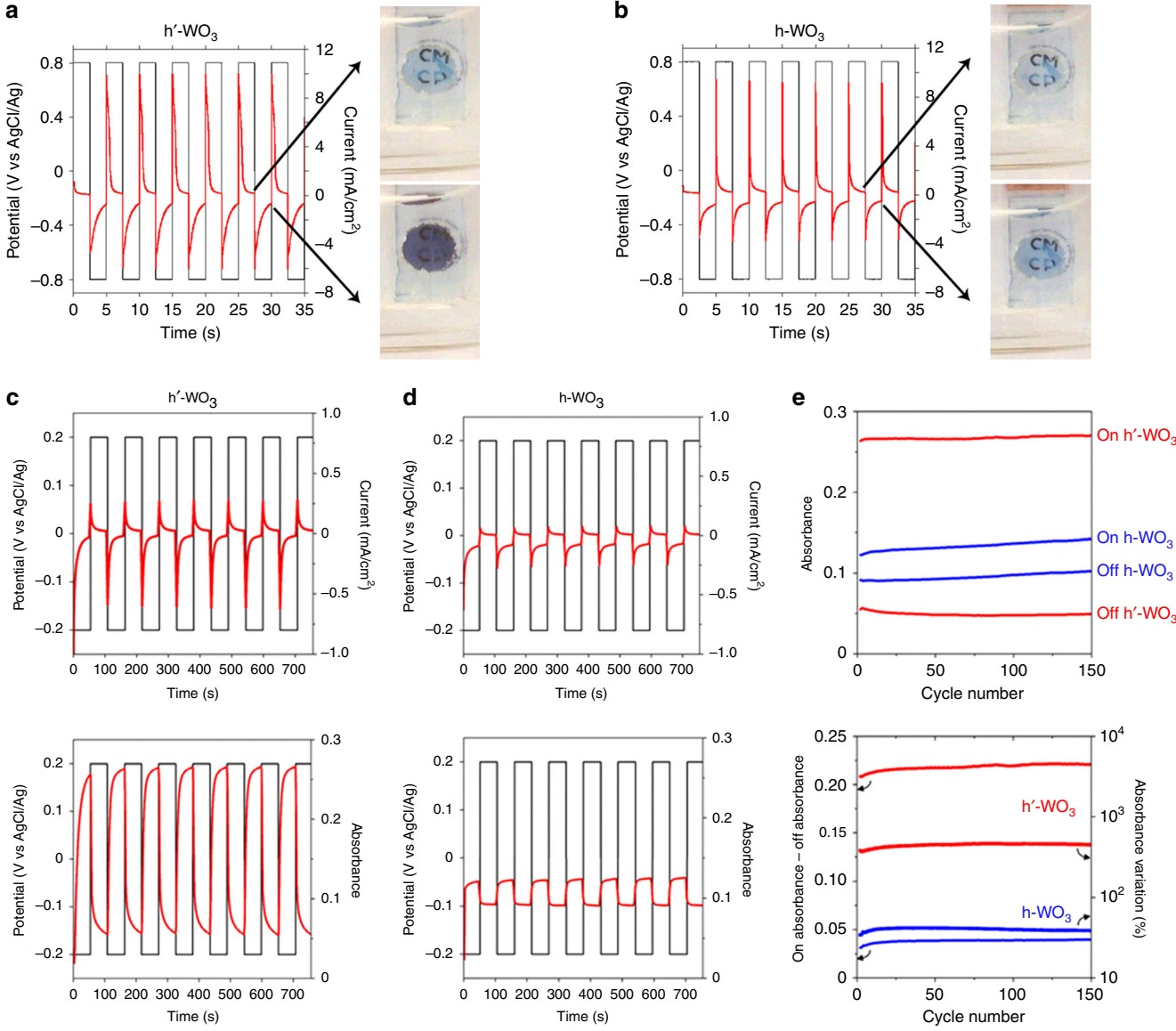

**Fig. 5** Electrochromic properties. Current evolution (red curves) upon application of a potential square function (black curves) at ±0.8 V vs. AgCl/Ag and 0.2 Hz for **a h'-WO₃** and **b** classical **h-WO₃** films spin coated on FTO substrates with H₂SO₄ 0.1 mol L⁻¹. Combined electrical and spectroscopic measurements of the electrochromic properties of **h'-WO₃** (**c**) and classical **h-WO₃** (**d**). Evolution of the current (top, red curves) and the corresponding in situ measured absorbance at 700 nm (bottom, red curves) during cycling between −0.2 and 0.2 V vs. AgCl/Ag at 0.009 Hz. **e** Variation of the absorbance upon cycling in the oxidized "off" (positive bias) and reduced "on" (negative bias) states for **h'-WO₃** and **h-WO₃**

containing additional four-octahedra channels. Besides the novelty of the crystal structure, the nanotexture is original and consists in 5 nm-thick nanoplatelets with the tunnels oriented perpendicularly to the basal face. The combination of microporosity with oriented nanostructure and high thermal and chemical stability is particularly suited to processes involving ions and molecules exchange and/or insertion. Among the large range of applications that can be foreseen, we have designed fast switching aqueous electrochromic devices by focusing on the ease and the rate of proton insertion. Compared with all previously known tungsten oxides, the new phase exhibits strong enhancement of the CE and coloration/decoloration cycling rate. Besides fast and stable electrochromics, this compound and its bronzes represent a realm of possibilities for water purification, separation, catalysis, electrocatalysis, energy conversion, and storage. These fields are currently under study. Finally, the stability of the novel W-O framework paves the way to a whole new family of compounds inspired from classical WO₃ phases, including alkali and ammonium bronzes. Isovalent and aliovalent tungsten substitutions

are also under investigation and might open the road to original nanomaterials with additional or improved functionalities.

## Methods

**Synthesis**. The hydrogen bronze of h'-WO₃ was prepared in hydrothermal conditions, as follows: sodium tungstate dihydrate, Na₂WO₄·2H₂O (Sigma) was dissolved in Milli-Q water to obtain a 0.15 mol L⁻¹ aqueous solution. 0.5 molar equivalents of hydrazine (molar ratio between hydrazine and W species) were added and the mixture was stirred for 1 h. Then the pH of the solution was adjusted to 0.6 with concentrated HCl (12 mol L⁻¹). Starting at pH 7, a white amorphous compound is formed and becomes beige below pH 1. The solution is then stirred at room temperature (or in a bath regulated between 25 and 30 °C) overnight to let the amorphous compound evolve. After that step, the reaction medium is heated at 120 °C in a closed borosilicate vial for 12 h. After 12 h, the reaction flask is cooled down to room temperature, the colorless supernatant is disposed of and the deep-blue powder is washed by centrifugation until the supernatant pH is neutral. A whitish deposit, corresponding to classical **h-WO₃** pollution is observed on the lining and must not be taken with the blue powder. Otherwise, it can be removed by centrifugated separation at 1000 rpm for 30 min. The sample is then dried at 40 °C under vacuum before grinding for further

characterization. The hydrogen bronze of **h'-WO₃** can also be obtained at 95 °C, by increasing the heating dwell time to 3 days.

**XRD and structure resolution**. The XRD patterns were recorded on a PanAlytical X'Pert Pro diffractometer equipped with a monochromator and using Cu Kα radiation. Details of the structure resolution are given in Supplementary information subsection "Structure resolution". The XRD reflections for the orthorhombic WO₃ structure were indexed along the ICDD file 04–007–2425.

**Electron microscopy**. SEM FESEM pictures were obtained using a Hitachi S3400-N operating at 5 kV. TEM images were recorded with a Tecnai spirit G2 apparatus operating at 120 kV. Atomic resolution characterization was performed on a JEOL JEMARM200cF electron microscope (Cold Emission Gun) operating at 80 kV provided with a spherical aberration corrector in probe (current emission density $\sim 0.7 \times 10^{-12}$ A cm$^{-2}$ and probe size $\sim 0.08$ nm), a GIF-QuantumER spectrometer and an Oxford INCA-350 detector. Atomic resolution STEM-HAADF and STEM-ABF images were acquired using inner and outer collection semiangles of 68 and 280 mrad, respectively, for HAADF and 11 and 22 mrad, respectively, for ABF with a nominal camera length of 60 mm. The presence of sodium and nitrogen was evaluated by recording EDS spectra in areas of 2–5 nm$^2$, with 50 s of total acquisition time.

**UV-visible-near infrared reflectance spectroscopy**. Absorption spectra between 200 and 2000 nm were calculated from total reflectance measurements on the compacted powder. These measurements were performed using a UV/Vis/NIR Perkin Elmer Lambda 900 with an integration sphere.

**X-ray photoelectron spectroscopy**. XPS has been performed at the Institut des Matériaux de Paris Centre on an OMicron apparatus operating at the incident energy of 1486.7 eV (Al K). Charge effects were minimized using a low energy electron beam. Powders were deposited on an indium substrate. Photoelectrons were collected with an angle of 90° with the sample surface. Measures were carried out with steps of 100 eV for the general spectra and 20 eV for the high-resolution spectra. The energy calibration was performed on the C 1-s peak, which possesses the lower energy (285 eV). The composition was then calculated using the Scofield photoemission effective section.

**Electron spin resonance spectroscopy**. The ESR and ferromagnetic resonance experiments were performed with a Bruker X-band spectrometer equipped with a SHQ resonator. The ESR spectra were lock-in detected with a modulation of the applied magnetic field at a frequency of 100 kHz and with a 10 Gauss modulation amplitude. The field offset of the electromagnet was measured with a Gaussmeter and the frequency was measured with microwave frequency counter. The samples were contained in quartz tubes. They were cooled down in a helium flow cryostat, which allowed varying the temperature between 4 K and 300 K. The samples could be photoexcited in situ with the unfiltered light from a HBO lamp or monochromatic light from an Ar ion laser.

**Ion exchange**. 34 mg of as-obtained powder were dispersed in 10 mL of a 1 mol L$^{-1}$ NaCl solution prepared with MilliQ water previously saturated with nitrogen by bubbling for 30 min. The suspension was kept under nitrogen flow. The pH then dropped quickly and stabilized after 15 min at a value of 2.9. This value was used to evaluate the amount of protons released and then the H$^+$/W ratio.

**Thermogravimetric analyses**. TGA/TDA experiments were performed on a Mettler LF1600 apparatus, under air and with a gas sweep of 30 mL min$^{-1}$. The temperature increase was controlled at 10 °C min$^{-1}$ between 25 and 1000 °C. Powder XRD patterns were recorded thanks to a thermal treatment chamber under air.

**Nitrogen sorption**. N₂-sorption for specific surface area measurements was performed on a BEL Belsorp Max apparatus. Prior to measurements, the powders were dried at 120 °C under vacuum overnight using a BEL Belprep II apparatus. The specific surface area was evaluated by using the Brunauer, Emmett, and Teller model.

**Modeling**. DFT calculations were performed with the Vienna ab initio simulation package (VASP)[56] using the Projected Augmented Wave Method (PAW)[57,58] applying the generalized gradient approximation (GGA). The GGA potential was developed using the PBE functional[59]. In a first step, all structural models were fully optimized using a plane wave energy cutoff of 550 eV. For h'-WO₃, h-WO₃, h'-WO₃ faulted, and monoclinic WO₃, we used respectively, 24, 12, 24, and 30 $k$ points in the irreducible Brillouin zone. All optimizations converged with residual Hellman–Feynman forces on the atoms smaller than 0.03 eV/Å and led to reasonable structures in comparison with experimental ones. Then, the optimized structures were used for accurate electronic structure calculations employing a plane wave energy cutoff of 550 eV and an energy convergence criterion of 10$^{-6}$ eV. Here, for **h'-WO₃**, **h-WO₃**, **h'-WO₃** faulted, and monoclinic **m-WO₃**, we used respectively 56, 90, 56, and 170 $k$ points in the irreducible Brillouin zone.

**Electrochemical measurements**. A 10 g L$^{-1}$ suspension of **h'-HₓWO₃** in ethanol was prepared by sonication during 1 h. Ten microliters were then deposited on an FTO substrates, which was spin coated at 2000 rpm for 10 s. This process was repeated five times. The same process was used to deposit classical **h-WO₃** films from nanorods of 10–15 nm in diameter and 100 nm in length, with a specific surface area of 45 m² g$^{-1}$ similar to the **h'-WO₃** sample. A surface of 0.785 cm² was selected by a scotch tape containing a hole tapped with a 5 mm diameter punch. In the photoelectrochemical experiments, the working electrode was immersed in the electrolyte with a surface of 1.1 cm² for h'-WO₃ and 1 cm² for h-WO₃. The transparent electrode was connected by copper tape and a wire to a three-electrode setup with saturated AgCl/Ag as reference and a Pt counter electrode. The electrolyte was H₂SO₄ 0.1 mol L$^{-1}$. A voltage square function was applied with a Solartron Modilab potentiostat between 0.8 and −0.8 V vs. AgCl/Ag or 0.2 and −0.2 V vs. AgCl/Ag, at a frequency of 0.1 or 0.009 Hz. The background and the reference for UV-visible measurements during cycling was a cell filled with the electrolyte and containing a FTO electrode without tungsten oxide film.

## Data availability

The data generated and analyzed during the current study are available from the corresponding author on reasonable request.

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

## Acknowledgements

This work was financially supported by Solvay Company, CNRS, and Sorbonne Université. The authors acknowledge D. Montero and C. Calers from the Institut des Matériaux de Paris Centre for SEM and XPS, respectively. They also acknowledge Patricia Beaunier from Laboratoire de Réactivité de Surface for TEM. Financial support from Research Project MAT2014-54372-R (MINECO, Spain) is also acknowledged.

## Author contributions

D.P. directed the study. J.B., B.M., S.C., and D.P. designed and interpreted the experiments for synthesis, characterizations and properties. J.B. and B.M. performed the synthesis and characterizations. B.M., D.P., and S.C. realized the electrochrome measurements. G.W. resolved the structure. H.J.V.B. and B.F. performed the ESR measurements, H.J.V.B., B.F. and C.S. interpreted the ESR data. H.K. performed DFT modeling. A.T.-P. and J.M.G.-C. performed the STEM study. V.B. and T.L.M. participated to the reflection on the results. D.P., J.B., B.M., S.C., H.K., and G.W. wrote the manuscript. All the authors discussed the results, commented on and revised the manuscript.

## Additional information

**Competing interests:** The authors declare no competing interests.

