## [Peer Review File · Nature Communications]

Reviewers' comments:

Reviewer #1 (Remarks to the Author):

The authors presented an interesting work that a novel OMS hexagonal polymorph of tungsten oxide (h'-WO₃) was successfully discovered for the first time. The material exhibits an unusual combination of 1-dimensional crystal structure and 2-dimensional nanostructure that ensures enhanced and fastened proton (de)insertion for stable electrochromic devices. I think this work is important to the inorganic functional materials. I recommend publication after minor revision.

1. The authors stated that "Protons are probably located into the (WO₆)₆ channels, but may also be present in the (WO₆)₄ channels, as already reported for lithium in the (WO₆)₄ openings of h-WO₃.⁴¹". However, there is no (WO₆)₄ in the structure of h-WO₃ (Figure 4).

2. h'-WO₃ is an actahedral molecular sieves material. The material was characterized via N₂ adsorption analysis. It is suggested to provide the N₂ isotherm adsorption-desorption curves of the sample. Furthermore, the authors should consider characterizing the microporosity of the sample via N₂ sorption analysis.

3. The electrochromic performance of h'-WO₃ sample is much better than that of h-WO₃ sample. The authors should give a detailed explanation.

4. An important article about the tungsten oxide materials should be cited.

Advanced Functional Materials, 2018, 28, 1707500.

Adv. Sci., 2018, 5, 1700986

Coordination Chemistry Reviews, 2018, 368, 13-34.

Advanced Functional Materials, 2017, 27, 1703949.

Reviewer #2 (Remarks to the Author):

In this manuscript, Besnardiere et al. report a novel OMS hexagonal polymorph of tungsten oxide (h' -WO₃) with fast exchange of ions and stability upon exchange. This discovery also may expand the WO₃ family. The topic is interesting although the synthetic method involved is not very novel, I have few comments about the data itself as I believe the authors were thorough with their analysis. The specific comments are as follows:

1. The authors should check the whole manuscript carefully, For example, on Page 5, "Figure 2e" (line 7) and "Figure 2f and g" (line 8) should be "Figure 1e" and "Figure 1f and g"
2. In Figure S3 and Figure S3, the coordinates should be adjusted clearer.
3. How to control the quantity of x in h' -H_xWO₃? As the authors introduced hydrogen in the (WO₆)₆ channels of their model in two different configurations (h' -H_{0.125}WO₃ and h' -H_{0.083}WO₃), can these be verified experimentally?
4. For the morphology of the as-prepared h' -WO₃, the authors described as "the powder is exclusively made of thin nanoplatelets between 50 and 100 nm and thickness of 5-10 nm". This is difficult to identify from Figure 1, besides, it should be better to provide data from statistical methods.
5. For comparison, the corresponding characterization of h-WO₃ should also be provided.
6. As an important means of proof for the identification of h' -WO₃ composition, EDS is not accurate enough since it is a semi-qualitative analysis. Other phase composition analysis, such as elemental analysis, XPS, XAS, ICP, etc. should be provided.
7. In Figure S13, there is clearly one pair of redox peak for h' -WO₃ while two pairs of redox peaks for h-WO₃, please give a brief explanation.

Reviewer #3 (Remarks to the Author):

The authors presented a work on h' -WO₃. They emphasised to use the term "octahedral molecular sieves (OMS)" and yet didnt show any tangible application for using this term in the whole paper

The depictions presented in Figures 1 to 4, in much more detail, have been presented in past works on WO₃. Many of the important references that include the assessment of various crystal phases and elemental arrangement of this materials have been deliberately removed from the references of this paper

Electrochromic property studies presented in this work are rudimentary and many main parameters are not presented

The response and recovery time of colouring are not even impressive in comparison to many of the past works

The OMS is fundamentally an important concept and could have been properly used in meaningful applications and relevant measurements could be included which are unfortunately not the case here

As such, I regretfully cannot support this paper for publication

Reviewer 1

The authors presented an interesting work that a novel OMS hexagonal polymorph of tungsten oxide (h' -WO₃) was successfully discovered for the first time. The material exhibits an unusual combination of 1-dimensional crystal structure and 2-dimensional nanostructure that ensures enhanced and fastened proton (de)insertion for stable electrochromic devices. I think this work is important to the inorganic functional materials. I recommend publication after minor revision.

We thank the reviewer for this comment.

1.1. The authors stated that “Protons are probably located into the (WO₆)₆ channels, but may also be present in the (WO₆)₄ channels, as already reported for lithium in the (WO₆)₄ openings of h -WO₃.⁴¹”. However, there is no (WO₆)₄ in the structure of h -WO₃ (Figure 4).

Yes, there are no (WO₆)₄ channels in the h -WO₃ structure, this is why we wrote “(WO₆)₄ openings”. These (WO₆)₄ openings are clearly observed on [100]* views of the h -WO₃ structure (and highlighted in ref 43 by Balaji et al.):

The same openings are actually also present in h' -WO₃: Figure 1g shows them explicitly through a [110] view.

1.2. h' -WO₃ is an octahedral molecular sieves material. The material was characterized via N₂ adsorption analysis. It is suggested to provide the N₂ isotherm adsorption-desorption curves of the sample. Furthermore, the authors should consider characterizing the microporosity of the sample via N₂ sorption analysis.

This has been added with new data in the revised manuscript page 14 by making reference to a **new Figure S15, showing the N₂ sorption isotherm of the h' -WO₃ phase but also of a h -WO₃ reference** for comparison. In the caption of this figure, we show that some microporosity is observed, as expected from crystal structures.

1.3. The electrochromic performance of h' -WO₃ sample is much better than that of h -WO₃ sample. The authors should give a detailed explanation.

To answer this comment, we added **characterizations of the classical h -WO₃ material in new Figure S17** for comparison (answering by the same way comment 2.6 by reviewer 2). **The discussion is added in the revised manuscript page 16:**

“The superior performances of the h' -WO₃ framework versus classical h -WO₃ in terms of response time and extent of coloration, hence of proton (de)insertion, may be ascribed to two features. First, additional 4-members (WO₆)₄ tunnels in the h' phase can accommodate protons as suggested by STEM-ABF images (**Figure S11**) and contribute to an increase in the coloration. Second, the original crystallographic orientation of the nanoplatelets provides optimal access to the tunnels of the h' phase while the channels of h -WO₃ are accessible only from the tips of the nanorods (**Figure S17**), with longer diffusion lengths for H⁺ (de)insertion, hence smaller (de)insertion rate.”

1.4. An important article about the tungsten oxide materials should be cited. (1) *Advanced Functional Materials*, 2018, 28, 1707500. (2) *Adv. Sci.*, 2018, 5, 1700986. (3) *Coordination Chemistry Reviews*, 2018, 368, 13-34. (4) *Advanced Functional Materials*, 2017, 27, 1703949.

Among the four references given by reviewer 1, references 2, 3 and 4 deal with iron oxides, manganese oxides, and Cu/Ag/Au sulfides. They are off topic. Reference 1 is indeed a review on tungsten-based materials for Li-ion batteries, including tungsten oxides. Our work does not deal with Li-ion batteries and the cited reference does not report our new WO₃ solid, so that there is no overlap at all.

To provide a recent reference on applications of WO_x-based materials for batteries, we included reference 1 of the reviewer in the revised manuscript as **reference 28 in the introduction, page 3.**

Reviewer: 2

In this manuscript, Besnardiere et al. report a novel OMS hexagonal polymorph of tungsten oxide (h'-WO₃) with fast exchange of ions and stability upon exchange. This discovery also may expand the WO₃ family. The topic is interesting although the synthetic method involved is not very novel, I have few comments about the data itself as I believe the authors were thorough with their analysis. The specific comments are as follows:

Precipitation of metal oxides is indeed "not very novel", but the specific protocol we apply is new, as it is the first one to yield this new WO₃ solid. What is needed for a precise control of polymorphism is a method that enables precise control over precipitation conditions. We show in this manuscript that aqueous synthesis in very soft conditions (95°C is enough, so the term "Soft chemistry" applies) meets these requirements and is, up to now, the only method to reach this new phase.

2.1. The authors should check the whole manuscript carefully, For example, on Page 5, "Figure 2e" (line 7) and "Figure 2f and g" (line 8) should be "Figure 1e" and "Figure 1f and g"

These mistakes have been corrected.

2.2. In Figure S3 and Figure S6, the coordinates should be adjusted clearer.

This has been corrected. These figures are now Figures S4 and S7.

2.3. How to control the quantity of x in h'-H_xWO₃?

According to electrochromic measurements presented in the manuscript, electrochemical (de)insertion makes it possible to adjust the quantity of protons in the h'-H_xWO₃. **This is clarified page 15 of the revised manuscript:**

"These color changes suggest that the amount of protons into the h'-H_xWO₃ bronze can be controlled by electrochemical (de)insertion."

2.4. As the authors introduced hydrogen in the (WO₆)₆ channels of their model in two different configurations (h'-H_{0.125}WO₃ and h'-H_{0.083}WO₃), can these be verified experimentally?

Following the same line, some authors (for instance Balaji et al. cited in the original manuscript, now as ref. 43 in the revised manuscript) have tried to specify experimentally the location of Li into h-Li_xWO₃ bronzes of the classical hexagonal phase. The study was very difficult and required a pre-established database of Raman spectra correlated to XRD patterns and lattice parameters changes as a function of the Li⁺ content. Such data are not available yet for our new phase. The problem is even trickier for proton bronzes, where the amount of protons is poor, as in our case. This is the reason why we turned to DFT modeling,

which has been already used to solve similar questions on other materials (see refs. 39 and 42 of the revised manuscript).

We have however considered carefully this interesting question. For this purpose, we turned back to STEM images with annular bright field detection, which provides a contrast sensitive to the light elements, including hydrogen in the best conditions (Ishakawa et al. Nat. Mater. 2011). Careful examination of Figure 1d (magnified below) and Figure 2e shows a dark contrast in all the $(\text{WO}_6)_6$ tunnels.

This contrast is not detected in the HAADF image so that it is due to a light element. Na is discarded as it is not detected by EDS (see the discussion below in our answer to comment 2.7). N is also discarded according to additional N-free syntheses (see page 9 of the manuscript). This contrast in ABF is then a confirmation that hydrogen atoms are in the $(\text{WO}_6)_6$ tunnels. Note that some of the $(\text{WO}_6)_4$ tunnels show also dark contrast (see below, arrows), which would suggest that some protons also occupy part of the $(\text{WO}_6)_4$ tunnels.

Finally, evaluating the position of the protons in the $(\text{WO}_6)_6$ channels is very difficult. Careful examination of the location of the dark contrast spots in the ABF image may suggest an off-center position, but the images are too noisy to provide an accurate measurement of the position of these spots.

These additional analyses and the full discussion on the composition are clarified in the revised manuscript with the following additions:

- **page 9:** “The presence of protons in the tunnels is confirmed by the STEM-ABF images (**Figure 1d** and **2e**) that show a dark contrast in all the $(\text{WO}_6)_6$ tunnels (**Figure 1d** inset) while HAADF (**Figure 1c**) does not exhibit such features, thus evidencing a light element: hydrogen atoms,³⁵ as N and Na have been discarded. Note that some of the $(\text{WO}_6)_4$ channels show also a dark contrast in ABF detection (**Figure S11**), which would suggest that some protons also occupy part of these 4-members tunnels.”

- **Modified Figure 1**, with inset showing contrast variations along the yellow line. Each large increase in intensity corresponds to each numbered $(\text{WO}_6)_6$ tunnel along the line. Black arrows show depressions of intensity in each tunnel.

- **New Figure S11.** STEM-ABF image of the h' - $\text{H}_{0.07}\text{WO}_3$ bronze showing dark contrast in all $(\text{WO}_6)_6$ channels and in some of the $(\text{WO}_6)_4$ channels (arrows). The dark contrast spots are not observed in STEM-HAADF mode and thus highlight the presence of a light element: hydrogen.

2.5. For the morphology of the as-prepared h'-WO₃, the authors described as “the powder is exclusively made of thin nanoplatelets between 50 and 100 nm and thickness of 5-10 nm”. This is difficult to identify from Figure 1, besides, it should be better to provide data from statistical methods.

This has been clarified, with **additional statistics** (thickness and basal width distributions), added as **new Figure S1** and **cited in the main text page 4**: “statistical measurements in SI”

2.6. For comparison, the corresponding characterization of h-WO₃ should also be provided

These data have been added as new **Figures S17**: XRD, TEM, HRTEM and the synthesis protocol.

2.7. As an important means of proof for the identification of h'-WO₃ composition, EDS is not accurate enough since it is a semi-qualitative analysis. Other phase composition analysis, such as elemental analysis, XPS, XAS, ICP, etc. should be provided.

We agree with reviewer 2's statement on EDS. That is why, in the original manuscript, we do not rely solely on EDS but we complement with additional characterization techniques. We understand our text was not clear enough.

First, XRD (page 8 of the revised manuscript) and the resulting electron density maps show a WO₃ framework with some additional atoms in the channels. Then, the most difficult questions deal with the identification and the quantification of these atoms. We have struggled a lot to provide a composition as accurate as possible:

We have not used ICP as the material is highly stable in acids (as shown in the electrochromism part) and then impossible to dissolve in these conditions, which are the most common for ICP. Instead, we have used other techniques:

Semi-quantitative analysis:

- EDS does not show any presence of Na.

quantitative analyses:

- Acid-base back-titration is efficient for OMS and molecular sieve materials. Exchange of the insertion cations with an excess of Na⁺ ions yields a release of H⁺ ions that corresponds to a H⁺/W ratio of 8 at.%. This shows that protons are inserted, this gives their content, but it is not discarding additional Na⁺ ions.
- XPS measurements are suggested by the reviewer but **they were actually already in the original manuscript (now in Figure S10)**. The W4f area yields a W5+/W ratio of 7 at.%. This is fully consistent with proton exchange. Note that XPS performed here probes ~10 nm thickness, which is enough to probe entire platelets. XAS would not provide any new data compared to those.

Overall, as suggested by reviewer 2, we have combined several phase composition analyses. All analyses are consistent with a composition where protons are the only inserted charge balancing cations in the h' bronze and we could quantify their amount. We also provide an evaluation of the error of this quantification. The composition of the deinserted phase h'-WO₃ is more straightforward: its blue color shows that it is only made of W⁶⁺, hence with a composition WO₃.

We have clarified these points by citing a new ref. (ref. 35) for proton imaging and by modifying the whole discussion on the composition page 9:

“Exchange of the inserted ions by an excess of Na⁺ in water (see Methods) yields fast acidification. Accordingly, the H⁺/W ratio is evaluated to 8 at. %. The additional presence of Na⁺ is unlikely as sodium could not be detected by energy dispersive X-ray spectroscopy (EDS,

Figure S9). This is confirmed by X-ray photoelectron spectroscopy (XPS, **Figure S10**) that yields a W^{5+}/W ratio of ca. 7 ± 3 at. %, in agreement with the proton content. Hence, sodium ions do not participate to charge balance and the bronze obtained after aqueous synthesis is a hydrogen bronze of overall composition $h'-H_{0.07 \pm 0.03}WO_3$. The presence of protons in the tunnels is confirmed by the STEM-ABF images (**Figure 1d** and **2e**) that show a dark contrast in all the $(WO_6)_6$ tunnels (**Figure 1d** inset) while HAADF (**Figure 1c**) does not exhibit such features, thus evidencing a light element: hydrogen atoms,³⁵ as N and Na have been discarded. Note that some of the $(WO_6)_4$ channels show also a dark contrast in ABF detection (**Figure S11**), which would suggest that some protons also occupy part of these 4-members tunnels."

2.8. In Figure S13, there is clearly one pair of redox peak for h' -WO₃ while two pairs of redox peaks for h-WO₃, please give a brief explanation.

This is an interesting question that could be answered by monitoring structural evolutions during electrochemical cycling (see comment 2.4). Such an arduous study is out of the scope of our manuscript. We can however speculate on the origin of this difference. Two pairs of redox peaks indicate two different sites of insertion, possibly two different channels, as reported by Balaji et al. for Li⁺ insertion in classical h-WO₃ (ref. 43). The single pair of redox peaks is probably due to strongly overlapping redox peaks with similar energetics and accessibility. **This discussion is added in the caption of revised Figure S18 (previously Figure S13):**

"Existence of one single pair of redox peaks for h' -WO₃, versus two pairs for h-WO₃, suggests that the different proton insertion sites in the h' phase are energetically equivalent or that H⁺ migration to these sites occurs at similar rates."

Reviewer: 3

The authors presented a work on h' -WO₃.

3.1. They emphasised to use the term "octahedral molecular sieves (OMS)". Yet didn't show any tangible application for using this term in the whole paper. The OMS is fundamentally an important concept and could have been properly used in meaningful applications and relevant measurements could be included which are unfortunately not the case here

We used the definition by S.L. Suib, who coined the term (*Acc. Chem. Res.* 2008, ref. 2 of the original manuscript): OMS are materials built from MO₆ octahedra where M is a transition metal. In the specific case of manganese OMS, manganese is in a mixed valence state and pores have diameter of ~0.46-0.69 nm. Our material fits in the definition: WO₆ octahedra, mixed valence of W when cations are inserted, and pore diameter ~0.51 nm.

We understand that reviewer 3 may have a concern with the literal meaning of octahedral molecular sieve, but consider that the term is widely accepted in the community, this is not the novelty claim. Overall, most works dealing with OMS focus on the combination of the various oxidation states that transition metals can bear with the ability to (de)insert cations. In the case of manganese, this has direct impact on properties for catalysis, sensing, and batteries, which are typically assessed in the OMS literature.

As the purpose of this manuscript is not to screen all possible properties of this new solid, we decided here to focus on aqueous **electrochromics: such devices are actually batteries relying on H⁺ (de)insertion (taking benefit from the microporosity), to which we add the ability of the material to change color upon insertion (versatility of oxidation states): this is a typical case of OMS properties.**

Besides, note that in the manuscript we also highlight the **ability of the solid to exchange H⁺ cations by Na⁺ cations when we titrate the amount of H⁺ (page 9): a second relevant measurement for assessing the OMS character.**

3.2. The depictions presented in Figures 1 to 4, in much more detail, have been presented in past works on WO₃. Many of the important references that include the assessment of various crystal phases and elemental arrangement of this materials have been deliberately removed from the references of this paper

We have serious concerns about the assumption of reviewer 3. Besides the fact that the reviewer questions our ethics without any evidence and in non-scientific terms, we note that he/she is not providing any example in the literature of the depictions he considers as not new.

We already cite some references about the various crystal phases of WO₃. The purpose of papers published in Nature Communications is, if we well understood, not to provide comprehensive literature reports, but to provide sufficient references in order to set the scientific context of our discovery, while demonstrating it is a discovery and why it is important.

Hence, to move on more scientific and constructive grounds than just assumptions, or belief, the underlying question of reviewer 3 is: how to ascertain that the WO₃ structure we report is new?

The answer is pretty simple: comprehensive structure databases (e.g. the ICSD and PDF databases) are not reporting this phase. The XRD pattern could not be indexed along any known solid, as already explained in the original manuscript page 4.

We understand that reviewer 3 would like to see some additional references on the crystal chemistry of WO₃. Ref. 17 was already present, but **we also have added the following references in the revised manuscript page 14**, which also set the basis for electrochromism:

- **ref. 49:** Granqvist, C. G. Electrochromic tungsten oxide: Review of progress 1993-1998. *Sol. Energy Mater. Sol. Cells* 60, 201–262 (2000).

- **ref. 50:** Niklasson, G. A. & Granqvist, C. G. Electrochromics for smart windows: Thin films of tungsten oxide and nickel oxide, and devices based on these. *J. Mater. Chem.* **17**, 127–156 (2007).

3.3. Electrochromic property studies presented in this work are rudimentary and many main parameters are not presented. The response and recovery time of colouring are not even impressive in comparison to many of the past works

Most works on electrochromic WO₃ films report the range of transmission change between the bleached and the colored states, the coloration efficiency, switching times, and stability upon cycling. We already reported most of these data in the original manuscript.

We mentioned response times without giving exact switching times (needed to reach 90% of the final absorbance), because they depend on many parameters as film processing, thickness and potential range that are different from one study to the other and make benchmarking difficult. One example: reviewer 3 mentions response/recovery times that “are not even impressive”. He/she is right for the values we mention in the text (~13s), corresponding to a +/- 0.2 V potential range (Figure 5c), which is narrow compared to many other works. But if we consider the +/- 0.8 V range (Figure 5a), closer to values used in many reports, we can evaluate roughly (with the current curves, although it should be rigorously evaluated on the absorbance curves) the switching times at 1.7s for coloring and 2.5s for bleaching. Those are much more “impressive” values, especially with non-optimized film processing.

This statement holds true for all other data mentioned (i.e. transmission change, coloration efficiency): because the film characteristics and the potential ranges are changing from one report to the other, we prefer not to claim (even if it is true for our conditions) very high performances, especially for our non-optimized films.

Concerning the parameters missing to characterize the films, indeed we had not provided the thickness of the films. **It is added in the revised manuscript page 15 with additional Figure S16 in SI:** “The film thickness was evaluated to 80-100 nm (Figure S16), which corresponds to approx. 8-20 stacked nanoplatelets.”

Stability over bleaching/coloring cycles was already demonstrated in Figure 5e and Figure S20. **We provide an additional data, showing by SEM that the nanostructuring of the film did not change after cycling. These data are shown in Figure S21.**

The purpose of the electrochromic property study is to highlight the impact and the potential benefits that could represent the new nanostructured solid we discovered. To do so, we focus on the comparison of the properties of structurally related phases, h' -WO₃ and h -WO₃. The comparison is meaningful because both samples are processed in the same way and with similar thickness while cycling parameters are identical.

This is the first time that h' -WO₃ is reported. We use electrochromic properties to point out the OMS character of the material (ion (de)insertion and oxidation state changes), with good performances in a non-optimized configuration. In our opinion, the importance and urgency of this manuscript lies in the discovery of a new functional nanostructured solid, which opens promising avenues for the development of many materials: optimized electrochromics but also battery electrodes, water splitting electrodes, and catalysis.

REVIEWERS' COMMENTS:

Reviewer #1 (Remarks to the Author):

It should be accepted now.

Reviewer #2 (Remarks to the Author):

The author has made a careful correction for this manuscript according to our suggestions. These responses to our questions are reasonable and satisfactory. As for some aspects unchanged, the author also gave a reasonable explanation. The overall level of this article has been greatly improved. So we recommend this article to publish in the journal of Nature Communications.

Reviewer #3 (Remarks to the Author):

The authors have not tried to be transparent about past works and ignored many of such works that comprehensively cover every bit of this whole paper

I recommend the rejection of the paper

The past work include

Hexagonal Tungsten Oxide Based Electrochromic Devices: Spectroscopic Evidence for the Li Ion Occupancy of Four-Coordinated Square Windows, Subramanian Balaji, Yahia Djaoued, André-Sébastien Albert, Richard Z. Ferguson and Ralf Brüning ,

Chem. Mater., 2009, 21 (7), pp 1381–1389

DOI: 10.1021/cm8034455

And many others